# No Geographical Inequalities in Survival for Sarcoma Patients in France: A Reference Networks’ Outcome?

**DOI:** 10.3390/cancers14112620

**Published:** 2022-05-25

**Authors:** Yohan Fayet, Christine Chevreau, Gauthier Decanter, Cécile Dalban, Pierre Meeus, Sébastien Carrère, Leila Haddag-Miliani, François Le Loarer, Sylvain Causeret, Daniel Orbach, Michelle Kind, Louis-Romée Le Nail, Gwenaël Ferron, Hélène Labrosse, Loïc Chaigneau, François Bertucci, Jean-Christophe Ruzic, Valérie Le Brun Ly, Fadila Farsi, Emmanuelle Bompas, Sabine Noal, Aurore Vozy, Agnes Ducoulombier, Clément Bonnet, Sylvie Chabaud, Françoise Ducimetière, Camille Tlemsani, Mickaël Ropars, Olivier Collard, Paul Michelin, Justine Gantzer, Pascale Dubray-Longeras, Maria Rios, Pauline Soibinet, Axel Le Cesne, Florence Duffaud, Marie Karanian, François Gouin, Raphaël Tétreau, Charles Honoré, Jean-Michel Coindre, Isabelle Ray-Coquard, Sylvie Bonvalot, Jean-Yves Blay

**Affiliations:** 1EMS Team–Human and Social Sciences Department, Centre Léon Bérard, 69008 Lyon, France; 2Research on Healthcare Performance RESHAPE, INSERM U1290, Université Claude Bernard Lyon 1, 69008 Lyon, France; 3ICR IUCT-Oncopole, 31000 Toulouse, France; chevreau.christine@iuct-oncopole.fr; 4Department of Surgical Oncology, Oscar Lambret Center, 59000 Lille, France; g-decanter@o-lambret.fr; 5Department of Clinical Research and Innovation, Centre Léon Bérard, 69008 Lyon, France; cecile.dalban@lyon.unicancer.fr (C.D.); sylvie.chabaud@lyon.unicancer.fr (S.C.); 6Department of Surgery, Centre Léon Bérard, 69008 Lyon, France; pierre.meeus@lyon.unicancer.fr (P.M.); francois.gouin@lyon.unicancer.fr (F.G.); 7Institut de Recherche en Cancérologie Montpellier, INSERM U1194, 34000 Montpellier, France; sebastien.carrere@icm.unicancer.fr; 8Service D’imagerie Diagnostique, Institut Gustave Roussy, 94800 Villejuif, France; leila.haddag-miliani@gustaveroussy.fr; 9Department of Pathology, Institut Bergonié, 33000 Bordeaux, France; f.le-loarer@bordeaux.unicancer.fr (F.L.L.); j.coindre@bordeaux.unicancer.fr (J.-M.C.); 10Centre Georges François Leclerc, 21000 Dijon, France; scauseret@cgfl.fr; 11Centre Oncologie SIREDO (Soins, Innovation et Recherche en Oncologie de l’Enfant, de l’aDOlescents et de L’adulte Jeune), Institut Curie, Université de Recherche Paris Sciences et Lettres, 75005 Paris, France; daniel.orbach@curie.fr; 12Radiologue, Département D’imagerie Médicale, Institut Bergonié, 33000 Bordeaux, France; m.kind@bordeaux.unicancer.fr; 13Department of Orthopaedic Surgery, CHU de Tours, Faculté de Médecine, Université de Tours, 37000 Tours, France; lr.lenail@chu-tours.fr; 14INSERM CRCT19 ONCO-SARC (Sarcoma Oncogenesis), Institut Claudius Regaud-Institut Universitaire du Cancer, 31000 Toulouse, France; ferron.gwenael@iuct-oncopole.fr; 15CRLCC Léon Berard, Oncology Regional Network ONCO-AURA, 69008 Lyon, France; helene.labrosse@rrcaura.fr (H.L.); fadila.farsi@onco-aura.fr (F.F.); 16Department of Medical Oncology, CHRU Jean Minjoz, 25000 Besançon, France; chaigneau.loic@orange.fr; 17Department of Medical Oncology, Institut Paoli-Calmettes, 13009 Marseille, France; bertuccif@ipc.unicancer.fr; 18CHU Réunion, 97410 Saint-Pierre, France; jean-christophe.ruzic@chu-reunion.fr; 19Service D’oncologie Médicale, CHU Dupuytren, 87000 Limoges, France; valerie.lebrun@chu-limoges.fr; 20Medical Oncology Department, ICO, 44800 Saint Herblain, France; emmanuelle.bompas@ico.unicancer.fr; 21UCP Sarcome, Centre François Baclesse, 14000 Caen, France; s.noal@baclesse.unicancer.fr; 22Department of Medical Oncology, Pitié Salpêtrière Hospital, Assistance Publique-Hôpitaux de Paris (AP-HP), Institut Universitaire de Cancérologie (IUC), CLIP(2) Galilée, Sorbonne University, 75013 Paris, France; aurore.vozy@aphp.fr; 23Centre Antoine-Lacassagne, 06000 Nice, France; agnes.ducoulombier@nice.unicancer.fr; 24Service d’Oncologie Médicale Hôpital Saint Louis, 75010 Paris, France; clement.bonnet@aphp.fr; 25Equipe EMS, Centre Léon Bérard, 69008 Lyon, France; francoise.ducimetiere@lyon.unicancer.fr; 26Service d’Oncologie Médicale, Hôpital Cochin, Institut du Cancer Paris CARPEM, Université de Paris, APHP Centre, 75014 Paris, France; camille.tlemsani@aphp.fr; 27INSERM U1016-CNRS UMR8104, Institut Cochin, Institut du Cancer Paris CARPEM, Université de Paris, APHP Centre, 75014 Paris, France; 28Orthopaedic and Trauma Department, Pontchaillou University Hospital, University of Rennes 1, 35000 Rennes, France; mickael.ropars@chu-rennes.fr; 29Département d’Oncologie Médicale, Hôpital Privé de la Loire, 42100 Saint-Etienne, France; olivier.collard@ramsaysante.fr; 30Service D’imagerie Médicale, CHU Hopitaux de Rouen-Hopital Charles Nicolle, 76000 Rouen, France; paul.michelin@chu-rouen.fr; 31Department of Medical Oncology, Strasbourg-Europe Cancer Institute (ICANS), 67033 Strasbourg, France; j.gantzer@icans.eu; 32Oncology Department, Centre Jean Perrin, 63011 Clermont-Ferrand, France; pascale.dubray-longeras@cjp.fr; 33Department of Medical Oncology, Cancer Institute of Lorraine-Alexis Vautrin, 54500 Vandoeuvre Les Nancy, France; m.rios@nancy.unicancer.fr; 34Department of Hepato-Gastroenterology and Digestive Oncology, Reims University Hospital, 51000 Reims, France; pauline.soibinet@reims.unicancer.fr; 35Medical Oncology, Insitut Gustave Roussy, 94800 Villejuif, France; axel.lecesne@gustaveroussy.fr; 36Department of Medical Oncology, CHU La Timone and Aix-Marseille Université (AMU), 13005 Marseille, France; florence.duffaud@ap-hm.fr; 37Department of Pathology, Lyon University Hospital, 69008 Lyon, France; marie.karanian@lyon.unicancer.fr; 38Medical Imaging Center, Institut du Cancer, 34000 Montpellier, France; raphael.tetreau@icm.unicancer.fr; 39Department of Surgical Oncology, Gustave Roussy, Villejuif 94800, France; charles.honore@gustaveroussy.fr; 40Department of Medical Oncology, Centre Leon Berard, 69008 Lyon, France; isabelle.ray-coquard@lyon.unicancer.fr; 41Department of Surgical Oncology, Institut Curie, Université Paris Sciences et Lettres, 75005 Paris, France; sylvie.bonvalot@curie.fr; 42Department of Medical Oncology, Centre Léon Bérard, Lyon University, 69008 Lyon, France; jean-yves.blay@lyon.unicancer.fr

**Keywords:** spatial inequalities, reference networks, remoteness, social deprivation, rare cancers, France

## Abstract

**Simple Summary:**

As patients with rare cancers face specific problems, reference networks have been developed in several European countries and then at the European level to improve their management. In some cases, the specialized centers belonging to reference networks provide remote services (specialized diagnosis review, discussion in the Multidisciplinary Tumour Board, etc.) to increase access to these services. Using data from the national sarcoma reference network implemented in France (NETSARC+), the IGéAS research program assesses the potential of its organization to address the geographical inequalities in cancer management. We analyze the individual, clinical, and geographical determinants of the overall survival of sarcoma patients in France. We found no association between the overall survival of sarcoma patients and variables measuring their social deprivation, remoteness from reference centers, and geographical context. Following previous results from the research program, this study suggests that reference network organization should be considered to reduce cancer inequalities.

**Abstract:**

The national reference network NETSARC+ provides remote access to specialized diagnosis and the Multidisciplinary Tumour Board (MTB) to improve the management and survival of sarcoma patients in France. The IGéAS research program aims to assess the potential of this innovative organization to address geographical inequalities in cancer management. Using the IGéAS cohort built from the nationwide NETSARC+ database, the individual, clinical, and geographical determinants of the 3-year overall survival of sarcoma patients in France were analyzed. The survival analysis was focused on patients diagnosed in 2013 (n = 2281) to ensure sufficient hindsight to collect patient follow-up. Our study included patients with bone (16.8%), soft-tissue (69%), and visceral (14.2%) sarcomas, with a median age of 61.8 years. The overall survival was not associated with geographical variables after adjustment for individual and clinical factors. The lower survival in precarious population districts [HR 1.23, 95% CI 1.02 to 1.48] in comparison to wealthy metropolitan areas (HR = 1) found in univariable analysis was due to the worst clinical presentation at diagnosis of patients. The place of residence had no impact on sarcoma patients’ survival, in the context of the national organization driven by the reference network. Following previous findings, this suggests the ability of this organization to go through geographical barriers usually impeding the optimal management of cancer patients.

## 1. Introduction

Cancer inequalities are a global challenge [1,2] that requires specific interventions to improve the outcomes of the most vulnerable populations [3]. The European atlas of cancer points out of the breadth of mortality inequalities among European countries [4], and these inequalities have been growing for decades in several developed countries [5,6,7]. Patients living in socially-deprived and rural areas are affected by worse survival outcomes that can be related either to their lower rate of referral or to a later referral to specialized cancer centres, with consistent results regardless of cancer type (colorectal, breast, lung, or prostate cancers) and study area, such as in Australia [8,9,10,11,12], in the USA [13,14,15,16], or in England [17,18]. In France, data from three cancer registries report the impact of the remoteness to the nearest cancer reference center on the survival of patients with colorectal cancer [17]. Others studies based on the registries data in France underline the influence of both social deprivation and remoteness to cancer reference centers on the access to specialized surgeons for patients with breast cancer [19], as well as on the access to reference centers for patients with colorectal cancer [20].

For patients with rare cancers, who have a worse survival than patients with common cancer, the lower accessibility of specialized facilities is particularly challenging and overlaps with other specific issues such as delays in diagnosis (due to less diagnostic precision) and therapeutic mismanagement [21,22]. Considering the value of the experience of the medical team on rare tumors for the patient outcome, reference networks have been implemented in several European countries to improve the management and survival of these patients [23,24]. According to the “hub-and-spoke” model, reference networks are designed as hierarchical organizations structuring collaborations between a relatively high number of centres (spokes) ensuring the spatial accessibility of cancer care and a limited number of reference centres (hubs) providing expert and highly-specialized services [23,25].

Sarcomas, which account for 1–3% of all cancers, are paradigmatic models for rare cancers. These tumors, which can be located anywhere in the body from connective tissue cells, are a heterogeneous group gathering dozens of histologic subtypes with heterogeneous clinical presentations and natural histories [26]. The estimation and trends over time of sarcoma incidence and survival are complicated by the lack of a unified method of reporting sarcomas, and also vary greatly according to histologic subtype [27]. Previous studies performed during the 2000s in France report that 30% of sarcomas are misclassified at initial diagnosis (sometimes mistaken for carcinomas) [28], and that only 40% to 50% of patients with sarcoma are treated according to the clinical guidelines for localized disease [29,30]. These results suggest a planned, coordinated, and specialized initial management in order to ensure the best possible management and survival [29,30,31,32]. The national sarcoma reference network NETSARC+ was launched in France in 2010, along with about 20 other networks accredited and supported by the French national cancer institute (INCa) to improve the management of rare cancer patients. The reference centers of the NETSARC+ network (Figure 1) provide sarcoma-specialized histological review and Multidisciplinary Tumor Board (MTB) discussion for each new sarcoma diagnosis, as required in the ESMO-EURACAN clinical practice guidelines [33]. Remote access to these specialized services can be delivered at the request of practitioners or facilities managing the patients.

As the NETSARC+ network is expected to improve the access to sarcoma expertise, the IGéAS research program was designed to assess the ability of this reference networks’ organization to address geographical inequalities in cancer management [22]. Previous analysis by the IGéAS research program showed the overriding impact of clinical factors (such as the type or size of tumor) on the access to NETSARC+ remote services in comparison to geographical variables. Indeed, the distance to reference centers had a slight influence on the access to these services, while social deprivation was not associated with access [34]. Here, we analyze the determinants of the survival of sarcoma patients within the French sarcoma reference network NETSARC+.

## 2. Materials and Methods

### 2.1. National Sarcoma Networks Databases

All patients with a review in a NETSARC+ center or discussed in the NETSARC+ sarcoma-specialized MTB were registered since 2010 in the NETSARC + database, gathering biological and clinical data (https://netsarc.sarcomabcb.org/, accessed on 19 April 2022). The database contains 60 items divided into four themes describing, for example, the characteristics of the patient (age, sex) and tumor (size, depth, grade, location, histologic type, and subtype), the diagnosis and review of the tumor, key information about the management (types and sites of first and potential secondary surgery, final quality of resection) and follow-up (relapse, survival), successive presentations of the file, and decision-making at MTB. The patients’ address, diagnosis and clinical data, place of surgery, and patient follow-up are prospectively collected and implemented in the database. The prospective collection and update of data is made from the electronic patient file of each NETSARC+ reference center by a dedicated onsite clinical research assistant. A quality assurance program has been established for these databases to ensure the quality of medical data recorded.

### 2.2. Constitution of the IGéAS Survival Cohort

The complete methodology of the IGéAS research program has been presented in a previous publication [22]. The inclusion criteria of the IGéAS cohort are:-Patient living in France at time of diagnosis;-Diagnosis of sarcoma/GIST/desmoid tumor/intermediate malignant tumor between 1 January 2011 and 31 December 2014;-Patient who benefiting from a review or a sarcoma MTB discussion in reference center belonging to the NETSARC+ network.

As patients’ follow-up and post-treatment data collection could not be conducted on the entire IGéAS cohort due to the large size of the cohort (N = 20589), our survival analysis focused on patients diagnosed in 2013. While the number of patients benefiting from the NETSARC+ network’s expertise has increased each year since its launch in 2010 [34], 2013 was the best compromise to ensure sufficient hindsight to collect patient follow-up. Due to their lower prognostic risk, intermediate malignancies, GISTs, and low-grade sarcomas were excluded from the survival study (Figure 2). As not all patients are managed in a sarcoma reference center in France, the various facilities that treated the patients were contacted in order to determine the vital status of the patients.

### 2.3. Statistical Analysis

The overall survival (OS) was defined as the time, expressed in months, between the date of diagnosis and the date of death or censored to the date of last news. OS was estimated with the Kaplan–Meier method and median follow up with the reverse Kaplan–Meier method. Cox proportional risk regression models (univariable and multivariable) were used to estimate the hazard ratios and their 95% confidence intervals.

The univariable analyses used individual variables (sex, age), clinical variables (tumor size, histological type and subtype, grade, stage, localization) and, thanks to the patient’s municipality of residence at diagnosis, some geographic indices previously published which measure the patient’s life context:-the GeoClasH classification distinguishing, through a K-means clustering, five types of French municipalities (metropolitan areas, precarious population districts, residential outskirts, agricultural and industrial plains, rural margins) from ten geographical scores measuring physical and social environments, as well as the spatial accessibility of health care [35];-the European Deprivation Index built by the ERISC platform (http://cancerspreventions.fr/inegalites-sociales/plateforme-2/, accessed on 24 May 2022) and based on ten social variables selected and weighted according to their association with individual data from the European Union Statistics on Income and Living Conditions survey [36];-the average travel time to the closest reference clinical center calculated with Odomatrix software [22].

All variables were included in a backward selection procedure to keep factors significant at a 5% level in the final multivariable model. All analyses were performed using SAS software, version 9.4 (SAS Institute, Cary, NC, USA).

## 3. Results

Our study included 16.8% bone, 69% soft-tissue, and 14.2% visceral sarcomas, with a median age of 61.8 years. The follow-up information needed for the survival study could be collected for 2281/2837 (80.4%) eligible patients. Patients without follow-up data and not included in the survival cohort were a little older (median of 65.6 years vs. 61.8 in the survival cohort), with smaller (33.4% with tumor less than 50 mm vs. 25.3% in the survival cohort), more superficial (25.2% vs. 16.6% in the survival cohort), and fewer grade three tumors (39% vs. 56.1% in the survival cohort). Patients without follow-up data and those included in the survival cohort were homogeneous in terms of social deprivation (European Deprivation Index), geographical context (GeoClasH), and spatial accessibility to reference centers.

Median follow up was 52.7 months (95% CI 50.8 to 53.6 months), and 781 deaths have been recorded among the 2281 patients included in the survival study. Seven variables were associated with the OS in the final multivariable model: sex, age, type of tumor, size of tumor, grade, internal trunk localization, and metastatic stage at diagnosis (Table 1). OS was better for female patients [HR = 0.86, 95% CI 0.74 to 1.0] and was worse for older patients, especially for patients over 70 years of age [HR = 4.29, 95% CI 2.64 to 6.97], as well as for patients with visceral sarcoma [HR = 1.33, 95% CI 1.01 to 1.75], with grade three sarcoma [HR = 1.78, 95% CI 1.48 to 2.14], with internal trunk sarcoma [HR = 1.49, 95% CI 1.23 to 1.80], and with metastatic stage at diagnosis [HR = 2.98, 95% CI 2.50 to 3.56].

No association with geographical variables was found. In univariable analysis alone, we found a significantly lower survival in precarious population districts [HR 1.23, 95% CI 1.02 to 1.48] in comparison to wealthy metropolitan areas (HR = 1), which may reflect a worse clinical presentation or a lower quality of management in precarious population districts. We estimated, in an additional OS model, the GeoClasH class hazard ratio after adjustment for the clinical variables to further this analysis. There was no longer any significant difference in survival between wealthy metropolitan areas (HR = 1) and precarious population districts [HR 1.03, 95% CI 0.85 to 1.25] after adjustment for the clinical variables (Table A1 in Appendix A). This result suggests that the clinical presentation of sarcoma patients living in wealthy metropolitan areas is more conducive to survival.

## 4. Discussion

Our study reports no geographical inequalities in the survival of patients with sarcoma, in the context of national organization driven by a reference network. This is an original result since inequalities in sarcoma management and survival have been found in the USA [37,38,39,40], in England [41], and in Denmark [42] based on national cancer databases or registries. While the clinical specificities of rare cancers and the reference centers’ remoteness could have contributed to strengthen the spatial inequalities in cancer management and survival [21], which have already been proven in France for many cancer locations [19,20,43], these results suggest the ability of the reference network organization to address the social and spatial inequalities in cancer management.

Indeed, a previous nationwide analysis of the determinants of early access to specialized services within the French sarcoma reference network NETSARC+ reports that, contrary to the overriding impact of some clinical factors, the distance to reference centers slightly alters the early access to sarcoma specialized services and social deprivation has no impact on it [34]. This ability is also illustrated by published maps showing the large geographical coverage of the French sarcoma reference centers that are often requested to review specimens or to discuss the therapeutic strategy of patients living several hundred kilometers away [22]. Combining clinical and geographical variables also makes it possible to understand the mechanisms leading to spatial inequalities in survival. Indeed, the survival inequalities observed in the univariable analysis according to the GeoClasH classes were related to the clinical presentation at time of diagnosis of sarcoma patients living in wealthy metropolitan areas, which is more conducive to survival, and could then not be attributed to a lower quality of management.

Our results complete and shed new light on another study observing no association between social deprivation and the survival of soft-tissue sarcoma patients conducted in the specific setting of a retrospective analysis in the single high-volume University of Washington Medical Center. According to this results, Eastman et al. suggest that “treatment at a high volume institution may mitigate the importance of socio-economic factors in the overall survival of soft-tissue sarcomas” [44]. Mandatory referral of all patients to reference centers is interesting to reduce inequalities in cancer management and survival, but here we found that such results can also be achieved through a more flexible organization driven by a reference network, following the model of hub and spoke, and which can help to avoid some potential adverse effects for patients related to the variable spatial accessibility of these reference centers [22] and the cumulative impact of long journeys on quality of life [45]. In France, practitioners or facilities managing sarcoma patients must request, at time of diagnosis, the remote specialized services (pathological review and MTB) of a reference center of the NETSARC+ reference network, following the ESMO-EURACAN clinical practice guidelines [33]. This early access (before first treatment) to sarcoma expertise should make it possible to tailor optimal treatment on a case-by-case basis within the framework of a collaboration between reference centers (hubs) and other facilities (spokes) [23,25]. As an example of this tailored network organization, sarcoma-specialized radiologists have designed and published, in the scope of the IGéAS research program, recommendations for radiological management which include referral recommendations according to the characteristics of the suspected soft-tissue tumor (https://expertisesarcome.org/prise-en-charge/radiologiep/, accessed on 24 May 2022) or bone tumor (https://expertisesarcome.org/espace-professionnels/suspicion-de-tumeur-osseuse/, accessed on 24 May 2022).

References networks have been implemented in France since 2010 to improve the quality of management and the survival of rare cancer patients. While the clinical (better compliance to international clinical guidelines, quality of initial management, and survival) and economic benefits of this organization have already been measured [31,32,46,47,48], assessing its potential effects on cancer inequalities was crucial, given the previous experience of efficient public health interventions that also led to an unforeseen increase in inequality. For example, universal programs to promote breast cancer screening have been successful in improving the screening uptake, but there are still “strong negative associations between screening uptake and area-level socio-economic deprivation” [49]. In the same manner, if the tobacco control policies have succeeded in reducing tobacco use, “the negative correlation between smoking prevalence and socioeconomic status has increased” [50]. The IGéAS research program has not observed any such “side effects” on inequalities for sarcoma patients since the implementation of the NETSARC+ reference network in France, either for inequalities in access to remote services [34], or for inequalities in survival thanks to this study.

To our knowledge, reference networks organizing sarcoma management have also already been implemented in Scandinavian countries as well as in the United Kingdom [51,52]. Moreover, three European Reference Networks (ERN) dedicated to rare cancers were launched in 2017: EuroBloodNet (https://www.eurobloodnet.eu, accessed on 24 May 2022), PaedCan (http://paedcan.ern-net.eu, accessed on 24 May 2022), and EURACAN (http://euracan.ern-net.eu, accessed on 24 May 2022). Each ERN gathers reference centers across Europe with highly skilled and multidisciplinary healthcare teams as well as advanced specialized medical equipment and infrastructures to ensure the optimal management of patients. As a limited number of facilities currently have the required expertise to suggest and deliver the optimal treatment to rare cancer patients, more and more patients with frequent cancers could also face this situation, given the scarcity and cost of the technical and human resources needed to implement precision medicine. In the context of an ongoing centralization of cancer care, ensuring the spatial accessibility of reference centers is critical, considering the potential side effects of patient remoteness on survival [18] and quality of life [45].

We recognize that our study also has some limitations or methodological biases. Even if the databases of the French sarcoma reference networks support an upwards reconsideration of the incidence of sarcomas [26], we cannot claim that our results are based on a nationwide exhaustive cohort of sarcoma patients. As an example, sarcoma-diagnosed patients without a pathological review or MTB discussion in reference centers cannot figure into our analyses. Given the estimated incidence of sarcoma in the international literature [53], we estimate the IGéAS cohort covers at least 90% of the national population. Moreover, 20% of the follow-ups were not collected for the 3-year survival analysis due to long-lost patients or non-response from the centers/practitioners contacted. The patients without follow-up were slightly older than those included in this study but otherwise have clinical characteristics (size, grade, depth of tumor) more conducive to survival, which allows us to rule out a selection bias in terms of initial prognosis. The median follow-up in the IGéAS survival cohort was 53 months but this varied greatly by region. The “region” variable was therefore excluded from the survival analysis because of this methodological bias. Considering the results’ interpretation, the more or less active traceability of reviews and MTB depending on the center could partly explain the regional variations observed. It may have been interesting to use social information at the individual level to complete the assessment of potential inequalities, but national sarcoma databases are not allowed to collect social information about patients. Finally, the use of the IRIS (infra-municipality) scale would have supported a more accurate measure of the social deprivation, but national sarcoma databases can only collect the patient’s municipality at diagnosis.

## 5. Conclusions

Place of residence is not associated with sarcoma patients’ survival in the context of national organization driven by a reference network. Following previous findings from the IGéAS research program, these results suggest the ability of this organization to push through the geographical barriers usually impeding the optimal management and degrading the prognosis of cancer patients. In order to ensure remote access to specialized services at a time when the development of precision medicine is leading to the spatial concentration of innovations in oncology, reference network organization should then be considered to reduce cancer inequalities.

## Figures and Tables

**Figure 1 cancers-14-02620-f001:**
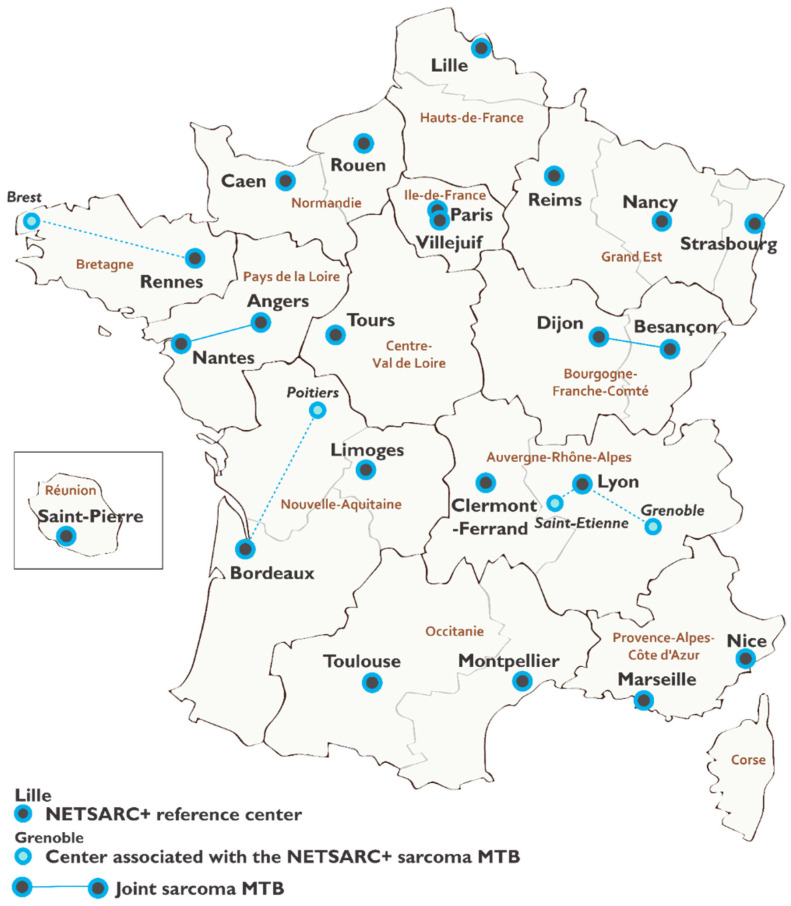
Mapping of the NETSARC+ reference network.

**Figure 2 cancers-14-02620-f002:**
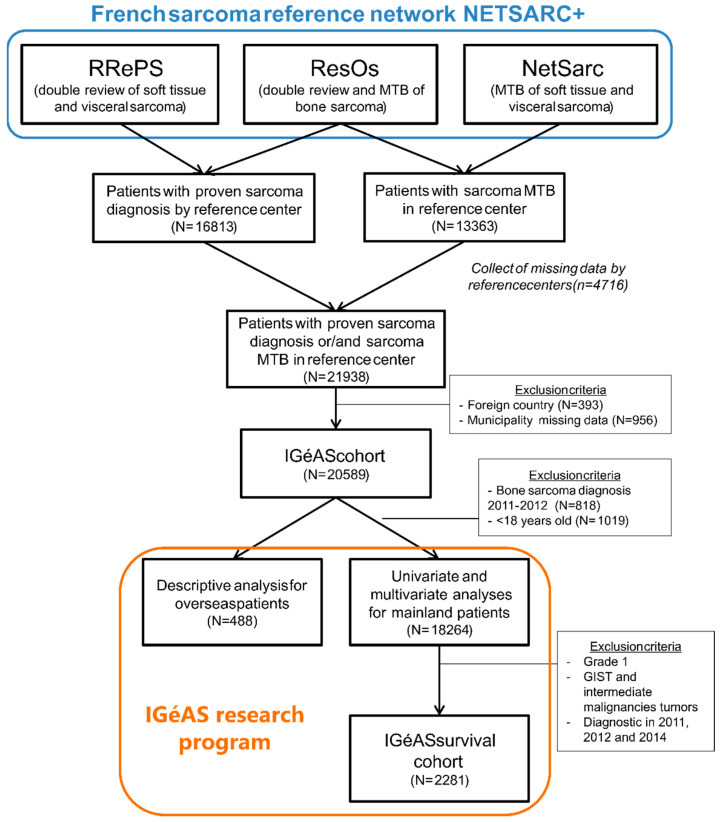
Flowchart of the IGéAS survival cohort.

**Table 1 cancers-14-02620-t001:** Clinical and geographical determinants of the 3-year death of sarcoma patients in France (source: IGéAS survival cohort, RRePS–ResOs–NETSARC databases).

Factor Label		Univariable (N = 2281)			Multivariable (N = 2152)	
	Deaths/N	Hazard Ratio (95% CI)	*p*-Value	Deaths/N	Hazard Ratio (95% CI)	*p*-Value
Sex			0.0363			0.0477
Male	429/1188	1		407/1120	1	
Female	352/1093	0.86 (0.75–0.99)		330/1032	0.86 (0.74–1.00)	
Age			<0.0001			<0.0001
[0–15[	21/97	1		19/89	1	
[15–18[	13/43	1.19 (0.60–2.37)		13/42	1.18 (0.58–2.40)	
[18–25[	36/97	1.88 (1.10–3.23)		35/95	2.00 (1.14–3.50)	
[25–50[	124/464	1.27 (0.80–2.02)		121/443	1.67 (1.02–2.73)	
[50–70[	298/838	1.98 (1.27–3.09)		282/795	2.64 (1.63–4.27)	
≥70	289/742	2.83 (1.82–4.41)		267/688	4.29 (2.64–6.97)	
Type of tumor			0.0007			0.0119
Bone	121/384	1		115/364	1	
Soft tissue	526/1573	1.21 (0.99–1.48)		494/1490	0.96 (0.76–1.21)	
Viscera	134/324	1.60 (1.25–2.05)		128/298	1.33 (1.01–1.75)	
Depth of tumor			0.0008			
Superficial	76/321	1				
Superficial and deep	42/144	1.19 (0.81–1.73)				
Deep	548/1470	1.51 (1.18–1.91)				
Missing	115/346	1.15 (0.86–1.54)				
Size of tumor			<0.0001			<0.0001
[0–50[	104/513	1		100/490	1	
[50–100[	227/687	1.61 (1.28–2.04)		217/661	1.46 (1.15–1.86)	
≥100	350/831	2.30 (1.84–2.86)		341/802	1.89 (1.50–2.38)	
Missing	100/250	2.91 (2.21–3.83)		79/199	1.88 (1.38–2.56)	
Grade			<0.0001			<0.0001
2	176/677	1		164/637	1	
3	515/1280	1.66 (1.40–1.97)		491/1215	1.78 (1.48–2.14)	
Missing	90/324	1.22 (0.95–1.58)		82/300	1.21 (0.93–1.59)	
Lower limb			0.0671			
No	641/1835	1				
Yes	136/439	0.84 (0.70–1.01)				
Upper limb			0.0153			
No	732/2095	1				
Yes	45/179	0.69 (0.51–0.93)				
Trunk wall			0.7757			
No	674/1968	1				
Yes	103/306	0.97 (0.79–1.19)				
Head and neck			0.1194			
No	731/2099	1				
Yes	46/175	0.79 (0.59–1.06)				
Internal trunk			<0.0001			<0.0001
No	588/1815	1		558/1718	1	
Yes	189/459	1.46 (1.24–1.72)		179/434	1.49 (1.23–1.80)	
Metastatic at diagnostic			<0.0001			<0.0001
No	542/1819	1		541/1817	1	
Yes	199/340	2.95 (2.50–3.47)		196/335	2.98 (2.50–3.56)	
GeoClasH classification of municipalities			0.2455			
Wealthy metropolitan areas	148/468	1				
Precarious population districts	414/1188	1.23 (1.02–1.48)				
Residential outskirts	108/321	1.20 (0.93–1.54)				
Agricultural and industrial plains	72/193	1.30 (0.98–1.72)				
Rural margins	39/111	1.23 (0.86–1.75)				
European Deprivation Index (quintiles)			0.8172			
≤−1.4 (least deprived)	159/462	1				
]−1.4; 1.7]	174/468	1.10 (0.89–1.37)				
]1.7; 5.5]	160/457	1.05 (0.84–1.31)				
]5.5; 8.8]	145/452	0.97 (0.77–1.21)				
>8.8 (most deprived)	143/442	1.02 (0.82–1.28)				
Travel time to the closest clinical sarcoma reference center (in minutes, quintiles)			0.8190			
≤30	158/467	1				
]30; 56]	153/440	0.97 (0.78–1.21)				
]56; 78.5]	163/451	1.09 (0.87–1.35)				
]78.5; 102]	147/467	0.96 (0.77–1.20)				
>102	160/456	1.02 (0.82–1.27)				

Hazard ratios were estimated with Cox proportional risk regression models.

## Data Availability

The data that support the findings of this study are available from the French Sarcoma Reference Network NETSARC+ but restrictions apply to the availability of these data, which were used under license for the current study, and so are not publicly available. Data are however available from the authors upon reasonable request and with the permission of the French Sarcoma Reference Network NETSARC+.

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
