# Peer review of "No Geographical Inequalities in Survival for Sarcoma Patients in France: A Reference Networks’ Outcome?"

_cancers, 2022, doi:10.3390/cancers14112620_

Round 1

Reviewer 1 Report

In this paper Fayet et al reported the results of a survival analysis in a cohort built in the frame of the French sarcoma network. The authors reported the effect on overall survival of variables measuring geographical and social aspects that could affect the management of sarcoma cases and ultimately survival.

Results did not show a relevant effect of these variables thus supporting indirectly the role of the network in reducing social inequalities in the management of these rare cancers.

The paper is interesting and well written I have only some points that should be addressed

- The Authors reported, among the limitations of the study 20% of lost to follow up. Please provide data on characteristics of these patients (ie age, place of residence, clinical aspect). and discuss of potential selection bias in relation to the specific analysis carried out

-More details on how vital status was obtained should be provided, in particular if the ascertainment was based on clinical data or obtained through linkage with mortality data archives.

-Although references for Geo ClasH classification and the European Deprivation index are provided, the Authors should add in the materials and methods section some information on how they are built

-Table 1 The table should be revised. In particular I suggest to add a footnote to indicate the models carried out to obtain univariate and multivariate estimates. Please check the labels of the quantitative variables, the reported ranges overlap

Author Response

R1:

In this paper Fayet et al reported the results of a survival analysis in a cohort built in the frame of the French sarcoma network. The authors reported the effect on overall survival of variables measuring geographical and social aspects that could affect the management of sarcoma cases and ultimately survival.

Results did not show a relevant effect of these variables thus supporting indirectly the role of the network in reducing social inequalities in the management of these rare cancers.

The paper is interesting and well written I have only some points that should be addressed

- The Authors reported, among the limitations of the study 20% of lost to follow up. Please provide data on characteristics of these patients (ie age, place of residence, clinical aspect). and discuss of potential selection bias in relation to the specific analysis carried out

Response 1 : We thank you for your review and comments. We provide additional data to compare the population used in this study and patients without follow-up data to rule out a potential selection bias in terms of initial prognosis.

-More details on how vital status was obtained should be provided, in particular if the ascertainment was based on clinical data or obtained through linkage with mortality data archives.

-Although references for Geo ClasH classification and the European Deprivation index are provided, the Authors should add in the materials and methods section some information on how they are built

Response 2 : In methods, we provide more information about the data collection as well as the geographical indices used in the study.

-Table 1 The table should be revised. In particular I suggest to add a footnote to indicate the models carried out to obtain univariate and multivariate estimates. Please check the labels of the quantitative variables, the reported ranges overlap

Response 3: The foot note for the table 1 is added but we don’t see the overlap for quantitative variables.

Reviewer 2 Report

Abstract  

  • It is unclear to me what is the type of study design (cohort or cross-sectional)? Why survival analysis was focused on patients diagnosed in 2013 only? (Line 83-84)
  • Need to state the age of patients and the types of sarcoma (e.g., bone, soft tissue..etc).

Keywords

  • Please add “France” to the list.

 Introduction  

  • Line 101-111: Specify where previous research discussed has been conducted. It would be useful to place the findings from previous research in the context of the country that the data comes from (particularly as the focus of the present study is French data). Please also specify the study design used and the types of cancer in these studies.
  • Authors may want to answer the following questions: What is sarcoma cancer causes, symptoms and treatments? What is the life expectancy with sarcoma? What are the chances of dying from sarcoma? What are the histological differences between carcinoma and sarcoma? How common is sarcoma in Europe, especially in France?
  • This paper lacks of scientific novelty/contribution or significance. Some studies related to the topic are missing (BMC Cancer. 2021;21(1):620; Asian Pac J Cancer Prev. 2014;15(1):25-8; Pediatr Blood Cancer. 2020; 67(12):e28708; Acta Oncologica 2020; 59(2): 127-133; JAMA Netw Open. 2020;3(8):e2011087). Authors should give specific reasons the importance of this study in light of other studies. This should be clearly stated in the last paragraph.

 Methods  

  • Insufficient description of individual, clinical and other factors. How these data were collected? This should be completed with more information.

 Discussion  

  • Line 211-229: This section is fine but needs to be strengthened and expanded. It is devoid of references to literature (Please refer to my comments in introduction). There is need to place the findings in a broad context.
  • Line 231-248: This paragraph should be moved to the end.

Author Response

R2:

Abstract  

  • It is unclear to me what is the type of study design (cohort or cross-sectional)? Why survival analysis was focused on patients diagnosed in 2013 only? (Line 83-84)

Response 1 : We thank you for your review, comments and supplemental references. We provide additional data and information in the manuscript to follow your all recommendations. This study is difficult to position in a specific field because we use data collected at a specific time (patients diagnosed in 2013), like a cross sectional study, but we have also a longitudinal approach since we are studying the survival of patients 3 years after their diagnosis. We have justified the choice of 2013 in the abstract.

  • Need to state the age of patients and the types of sarcoma (e.g., bone, soft tissue..etc).

Response 2 : Done

Keywords :  

  • Please add “France” to the list.

Response 3 : Done

 Introduction  

  • Line 101-111: Specify where previous research discussed has been conducted. It would be useful to place the findings from previous research in the context of the country that the data comes from (particularly as the focus of the present study is French data). Please also specify the study design used and the types of cancer in these studies.
  • Authors may want to answer the following questions: What is sarcoma cancer causes, symptoms and treatments? What is the life expectancy with sarcoma? What are the chances of dying from sarcoma? What are the histological differences between carcinoma and sarcoma? How common is sarcoma in Europe, especially in France?
  • This paper lacks of scientific novelty/contribution or significance. Some studies related to the topic are missing (BMC Cancer. 2021;21(1):620; Asian Pac J Cancer Prev. 2014;15(1):25-8; Pediatr Blood Cancer. 2020; 67(12):e28708; Acta Oncologica 2020; 59(2): 127-133; JAMA Netw Open. 2020;3(8):e2011087). Authors should give specific reasons the importance of this study in light of other studies. This should be clearly stated in the last paragraph.

Response 4 : Done

 Methods  

  • Insufficient description of individual, clinical and other factors. How these data were collected? This should be completed with more information.

Response 5 : In methods, we provide more information about the data collection

 Discussion  

  • Line 211-229: This section is fine but needs to be strengthened and expanded. It is devoid of references to literature (Please refer to my comments in introduction). There is need to place the findings in a broad context.
  • Line 231-248: This paragraph should be moved to the end.

Response 6 : Done, thank you for you recommandations

Reviewer 3 Report

The authors present a retrospective study demonstrating that there is no correlation between the geographical location of sarcoma incidence and patient survival. The study is interesting because it is generally assumed that regions characterized by lower income would have worse patient prognosis. However, the caveat is the limited samples size and particularly it is focused only in the patients from 2013. Would it be possible to add data from another year, to rule out bias?

Best wishes

Author Response

R3:

The authors present a retrospective study demonstrating that there is no correlation between the geographical location of sarcoma incidence and patient survival. The study is interesting because it is generally assumed that regions characterized by lower income would have worse patient prognosis. However, the caveat is the limited samples size and particularly it is focused only in the patients from 2013. Would it be possible to add data from another year, to rule out bias?

Response 1 :We thank you for your review and comments. We have justified the choice of 2013 in the abstract and provide additional data to compare the population used in this study and patients without follow-up data to rule out a potential selection bias in terms of initial prognosis.

Reviewer 4 Report

thank you for giving me the opportunity to review the paper

the authors report the results of a national cohort and were interested in the impact of non-clinical determinants.

Were the operating volume and the number of patients treated in each centre assessed in the results?

Was there a relationship between the deprivation and the stage at discovery of the disease?

Author Response

R4 :

Comments and Suggestions for Authors

thank you for giving me the opportunity to review the paper

the authors report the results of a national cohort and were interested in the impact of non-clinical determinants.

Were the operating volume and the number of patients treated in each centre assessed in the results?

Response 1 : We thank you for your review and comments. No, they were not assessed in this study, dedicated to the measure of spatial inequalities, because the impact of care in reference centers (which have the highest volume of patients) has already been demonstrated in previous studies (https://pubmed.ncbi.nlm.nih.gov/31081028/, https://pubmed.ncbi.nlm.nih.gov/31065964/).

Was there a relationship between the deprivation and the stage at discovery of the disease?

Response 2 : We have not tested the relationship between the deprivation and the stage at diagnosis in this study but it has always been reported by many studies involving different types of cancers and countries e.g.:

  1. Rutherford MJ, Ironmonger L, Ormiston-Smith N, Abel GA, Greenberg DC, Lyratzopoulos G, et al. Estimating the potential survival gains by eliminating socioeconomic and sex inequalities in stage at diagnosis of melanoma. Br J Cancer. 2015 Mar 31;112 Suppl 1:S116-123.
  2. Auluck A, Walker BB, Hislop G, Lear SA, Schuurman N, Rosin M. Socio-economic deprivation: a significant determinant affecting stage of oral cancer diagnosis and survival. BMC Cancer. 2016 02;16:569.
  3. Belot A, Fowler H, Njagi EN, Luque-Fernandez MA, Maringe C, Magadi W, et al. Association between age, deprivation and specific comorbid conditions and the receipt of major surgery in patients with non-small cell lung cancer in England: A population-based study. Thorax. 2019 Jan 1;74(1):51–9..

Round 2

Reviewer 1 Report

My suggestion is "Accept in the present form"

Reviewer 2 Report

No further comments.

Reviewer 3 Report

The authors have addressed my concerns. I recommend acceptance of the manuscript.

Best wishes